# FOCUS: A FREQUENCY-ORIENTED AND CLASS-UNDERREPRESENTED SEMANTIC SEGMENTATION FRAMEWORK FOR FOOD IMAGES

## ABSTRACT

Generating high-quality semantic segmentation results from food images remains a challenging task, particularly in the presence of complex boundaries and class imbalance. Existing methods often struggle with blurred edges and underperform on long-tailed categories, limiting their generalizability in practical scenarios. To address these issues, we propose FOCUS, a novel semantic segmentation framework designed to enhance boundary precision and improve underrepresented class recognition. Specifically, we introduce a frequency-based strategy that selectively processes high-frequency components via differential convolution and integrates explicit edge supervision during training. This enables the model to better capture fine-grained boundary details and improves edge discriminability. To mitigate class imbalance, we introduce an enhanced gradient allocation mechanism that applies targeted matching supervision to underrepresented categories, thereby amplifying learning signals for low-shot classes and improving classification accuracy. Extensive experiments on benchmark datasets, FoodSeg103, UECFood-PixComplete, and Food50Seg, show that FOCUS consistently outperforms existing approaches in both boundary quality and underrepresented class performance, validating its architectural effectiveness and robust generalization capability.

## 1 INTRODUCTION

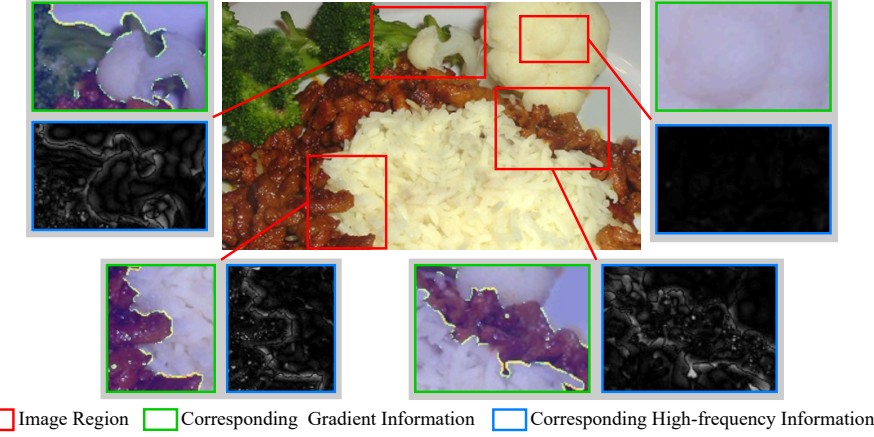

Figure 1: Gradient and frequency representation in food images. The green box shows gradient information, while the blue box highlights high-frequency components from the Fourier transform, with edges exhibiting abundant high-frequency content and high gradients.

Semantic segmentation of food images is a fundamental task in food computing, enabling various downstream applications such as food quality assessment Zhang et al. (2021b) and nutritional analysis He et al. (2020). Early approaches primarily relied on pixel-wise classification and performed

well when high-quality pixel-level annotations were available. However, in the food domain, such fine-grained annotations are often scarce, with datasets typically providing only coarse polygonal labels, which poses significant challenges for conventional methods. To overcome these limitations, recent research has increasingly adopted the mask classification paradigm Cheng et al. (2021), which directly predicts object-level masks. This strategy not only improves boundary localization but also enhances category recognition, thereby achieving higher-quality food image segmentation Min et al. (2019).

Despite recent advances, the inherent visual complexity and high intra-class variability of food items continue to present significant challenges for achieving robust segmentation. One major obstacle is the low quality of mask proposals near object boundaries. This issue arises primarily from the irregular and diverse shapes of food items Lin et al. (2017), as well as boundary interference caused by common food presentation styles such as overlapping and stacking. To address this, prior works have explored supervised attention mechanisms Chen et al. (2020) to enhance boundary awareness and refine mask predictions. However, these approaches typically rely on standard convolutional operations, which do not explicitly model boundary-specific structures. Because traditional convolutions operate over broad and unconstrained receptive fields, they often fail to effectively capture fine-grained edge details, resulting in suboptimal boundary localization. Therefore, improving mask proposal accuracy in boundary regions remains critical for advancing segmentation quality in food image analysis.

Another critical challenge observed in real-world scenarios is the pronounced class imbalance Cui et al. (2019), where certain categories are represented by only a limited number of training samples. These rare classes pose greater difficulties for model optimization, necessitating more effective strategies to fully exploit the available data during training. In food image generation, rare categories such as unique dishes or uncommon ingredients, are typically underrepresented, resulting in limited supervision and sparse gradient updates. Consequently, models struggle to learn stable and discriminative feature representations. As a result, the accuracy of mask classification for these classes is significantly compromised. Prior studies have attempted to mitigate class imbalance through approaches such as data augmentation Ghiasi et al. (2021) and loss re-weighting Wang et al. (2021). While these methods offer some improvement, they generally rely on explicit adjustments to the loss function or sample distribution and fail to provide richer and more diverse gradient signals. However, they rarely explore the optimization process from an algorithmic perspective. Consequently, they often fall short of fundamentally addressing the low mask classification accuracy associated with underrepresented classes.

To address aforementioned challenges of boundary ambiguity and underrepresented class learning, we propose a novel framework, FOCUS, which enhances both mask proposal quality and mask classification, thereby improving overall segmentation accuracy. From a frequency-aware perspective ( Figure 1), boundary regions exhibit sharp gradient transitions and abundant high-frequency components, reflecting the complexity of object edges. To effectively capture these signals, we introduce a Frequency-Aware Boundary Modeling mechanism that replaces standard convolution with differential convolution to explicitly incorporate edge priors. And this design employs a supervised attention mechanism to capture subtle boundary variations while balancing high- and low-frequency information, thereby generating more accurate masks. To further improve classification performance for underrepresented categories, we introduce an extra matching strategy that assigns multiple queries to each foreground object, thereby expanding their effective receptive fields without modifying the overall training paradigm (Figure 2). This strategy provides more diverse gradient signals, enabling stronger supervision and more stable learning for these underrepresented classes.

Our main contributions are as follows:

- We propose FOCUS, a novel mask classification-based semantic segmentation framework tailored for food images. FOCUS effectively addresses the challenges of food image segmentation and achieves state-of-the-art (SOTA) performance on multiple benchmark datasets, as validated by extensive ablation studies and qualitative results.

- We present a new edge representation paradigm that combines frequency-domain selective filtering with differential convolution. This design decouples fine-grained edge signals from background noise, enabling more accurate boundary modeling and addressing long-standing difficulties in capturing intricate object boundaries.

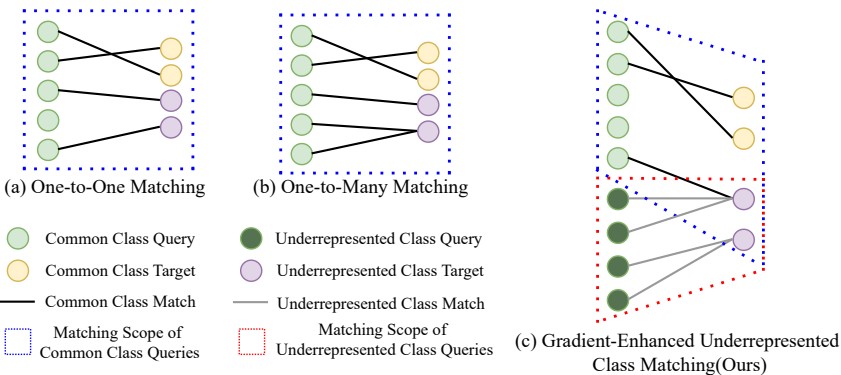

Figure 2: A comparison between existing matching strategies, including One-to-One Matching and One-to-Many Matching, and the Gradient-Enhanced Underrepresented Class Matching (Ours)

- We introduce a gradient-enhanced matching strategy that dynamically allocates extra queries to underrepresented and critical categories. This strategy improves classification accuracy and stabilizes training under class imbalance, thereby strengthening the overall effectiveness of the mask segmentation pipeline.

## 2 METHOD

### 2.1 OVERVIEW

Figure 3 illustrates the overall architecture of the proposed FOCUS framework, which consists of three main components: (1) the foundational backbone and pixel decoder for feature extraction, (2) the Frequency-Aware Boundary Modeling (FABM) that enriches boundary representations in critical feature maps, and (3) the Gradient-Enhanced Underrepresented Class Matching (GEUCM), which allocates extra queries to underrepresented classes to enhance classification accuracy.

### 2.2 FREQUENCY-AWARE BOUNDARY MODELING

Boundary regions typically exhibit sharp gradient transitions and rich high-frequency components, reflecting the inherent complexity of object edges. To effectively capture these signals, we design a Frequency-Aware Boundary Modeling (FABM) module, inspired by concepts from FADC Chen et al. (2024a) and DEA-Net Chen et al. (2024b). However, unlike the FADC and DEA-Net methods that perform fine-grained frequency decomposition, our approach simplifies frequency division into high and low frequencies, processing only the high-frequency components for further refinement. FABM primarily consists of a Frequency-Differentiated Block (FDB) and a convolutional layer.

**Frequency Difference Block** FDB primarily comprises two components: frequency-based feature selection and difference convolution. By isolating high-frequency signals and capturing pixel-level changes through difference-structured kernels, FDB effectively combines local detail sensitivity with global structural awareness for robust edge modeling.

Specifically, given an input feature map $F_{in}$, frequency-based feature selection first employs the Fast Fourier Transform (FFT) to map it from the spatial domain to the frequency domain. Then, based on a predefined frequency threshold $\alpha$, two binary masks are constructed to represent the low-frequency and high-frequency components of the input feature, respectively. Finally, the frequency-domain feature is element-wise multiplied with these two masks and projected back into the spatial domain via the Inverse FFT, yielding the high- and low-frequency feature representations divided by the threshold.

$$F_{\text{low}} = \mathcal{F}^{-1}(M_{low}\mathcal{F}(F_{in})), \quad F_{\text{high}} = \mathcal{F}^{-1}(M_{high}\mathcal{F}(F_{in})), \tag{1}$$

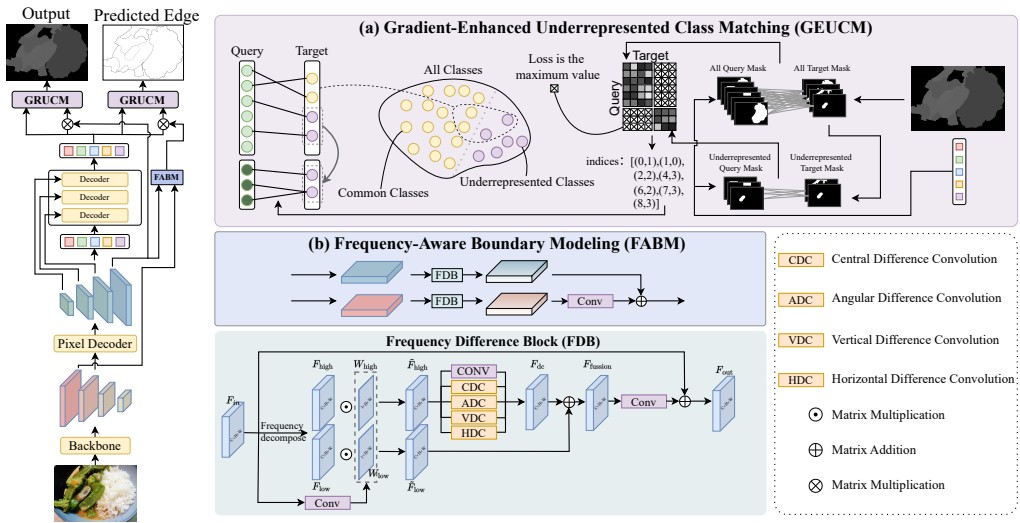

Figure 3: Architectural diagram of the proposed FOCUS framework. Given an input image, our model extracts multi-level features through a backbone and pixel decoder. These features, together with learnable queries, are fed into a Transformer decoder to produce the final segmentation output. To enhance the mask proposal quality and mask classification accuracy, we accordingly introduce two key modules: (a) FABM, which strengthens boundary representations in critical feature maps, and (b) GEUCM, which allocates additional queries to underrepresented classes to boost classification accuracy.

$$M_{\text{low}}(u, v) = \begin{cases} 1, & \max(|u|, |v|) < \alpha \\ 0, & \text{otherwise} \end{cases}, \qquad M_{\text{high}}(u, v) = 1 - M_{\text{low}}(u, v) \qquad (2)$$

Here, $\mathcal{F}$ and $\mathcal{F}^{-1}$ denote the FFT and inverse FFT, respectively. $M_{\text{low}}$ and $M_{\text{high}}$ are binary masks used to extract the corresponding low- and high-frequency components, and $u$ and $v$ indicate the frequency coordinates along the width and height dimensions. The resulting $F_{\text{low}}$ and $F_{\text{high}}$ represent the low-frequency and high-frequency spatial-domain features, respectively.

Next, a learnable convolutional layer is applied to selectively retain or suppress specific frequency components. By convolving the input feature $F_{in}$ with this convolutional layer, two weighting maps, $W_{\text{low}}$ and $W_{\text{high}}$, are generated to modulate the low-frequency and high-frequency features, respectively.

$$W_{\text{low}} = \text{Conv}_{\text{low}}(F_{in}), W_{\text{high}} = \text{Conv}_{\text{high}}(F_{in}). \qquad (3)$$

$$\tilde{F}_{\text{low}} = W_{\text{low}} \odot F_{\text{low}}, \tilde{F}_{\text{high}} = W_{\text{high}} \odot F_{\text{high}}. \qquad (4)$$

Here, $W_{\text{low}}, W_{\text{high}} \in \mathbb{R}^{1 \times H \times W}$ are represent spatial selection maps learned from the input feature $F_{in}$ through $\text{Conv}_{low}$ and $\text{Conv}_{high}$, respectively. The weighted low- and high-frequency features are denoted as $\tilde{F}_{\text{low}}$ and $\tilde{F}_{\text{high}}$, respectively.

The weighted high-frequency feature $\tilde{F}_{\text{high}}$ is further enhanced through a series of difference convolutions, which strengthen edge and fine-grained details, producing the feature $F_{\text{dc}}$.

$$F_{\text{dc}} = \text{DConv}(\tilde{F}_{\text{high}}), \qquad (5)$$

The enhanced high-frequency feature is then fused with the weighted low-frequency feature $\tilde{F}_{\text{low}}$ through element-wise addition to obtain the fused representation $F_{\text{fusion}}$.

$$F_{\text{fusion}} = F_{\text{dc}} + \tilde{F}_{\text{low}}. \qquad (6)$$

Finally, $F_{\text{fusion}}$ is further refined via a standard convolutional layer and combined with the original input feature $F_{in}$ to produce the final output $F_{out}$.

$$F_{out} = \text{Conv}(F_{\text{fusion}}) + F_{in}. \tag{7}$$

**Difference Convolution** Difference convolution employs differential kernels to highlight local pixel variations, making it effective for capturing gradients, edges, and structural details Yu et al. (2021); Su et al. (2021). Different kernel configurations (e.g., horizontal, vertical, angular, and central) emphasize gradient changes along specific directions Yu et al. (2020); Chen et al. (2024b). Unlike prior studies that mainly focus on designing new variants, our work integrates these operations with frequency-based feature selection in the FABM module, enabling explicit boundary enhancement for segmentation.The difference convolution can be formulated as follows:

$$Y(i,j) = \sum_{k=-1}^{1} \sum_{l=-1}^{1} I(i+k, j+l) \cdot W(k,l), \tag{8}$$

where $Y(i,j)$ is the pixel value of the output image after difference convolution, $I(i+k, j+l)$ is the pixel value of the input image at the position $(i+k, j+l)$, and $W(k,l)$ is the weight of the kernel at the corresponding position. In difference convolution, the kernel is designed with a differential structure. For example, in a horizontal $3 \times 3$ convolution, only the left and right columns contain learnable parameters, the middle column is fixed to zero, and the parameters in the same row are constrained to be negatives of each other. This design highlights pixel differences along the horizontal direction, thereby enhancing gradients and edge features.

Formally, the kernel can be expressed as:

$$W = \begin{bmatrix} -a_1 & 0 & a_1 \\ -a_2 & 0 & a_2 \\ -a_3 & 0 & a_3 \end{bmatrix},$$

where $a_1, a_2, a_3$ are learnable parameters.

**FABM Loss Design** To further enhance the model's ability to capture object contours, we introduce an additional boundary prediction branch alongside the original mask classification loss. The loss is defined as $L_{\text{FABM}} = L_{\text{edge}} + \lambda_{cls} L_{cls}$, where all loss terms in $L_{\text{FABM}}$ are computed using the predicted and ground-truth edge maps. The boundary supervision follows the same binary cross-entropy and Dice formulation used for mask prediction: $L_{\text{edge}} = \lambda_{ce} L_{ce} + \lambda_{dice} L_{dice}$. By explicitly supervising boundary information, this branch encourages the model to focus on precise contour details, leading to clearer boundary representations and improved overall segmentation performance.

## 2.3 Gradient-Enhanced Underrepresented Class Matching

The following section introduces the definition of underrepresented classes specific to our task, and then elaborates on the design of the Gradient-Enhanced Underrepresented Class Matching module.

**Definition of Underrepresented Class** In real-world food image datasets, category distributions are typically imbalanced, with certain classes appearing rarely or consistently yielding poor segmentation performance. We collectively refer to these as *underrepresented classes* to facilitate targeted optimization.

To formalize this, consider a dataset with $n$ food categories. We begin by computing the occurrence frequency of each category, defined as the number of training images in which the category appears divided by the total number of training images. We then identify the $k$ least frequent categories as a candidate underrepresented class set. From this set, we exclude $j$ categories that demonstrate strong model performance on a validation set, those whose IoU or accuracy exceed a predefined threshold $m$. The remaining $k - j$ categories form Category Set A. Next, we evaluate segmentation performance on the remaining $n - k$ categories and select the $j$ worst-performing ones based on the same metric, forming Category Set B. The final *underrepresented class* set is defined as the union of Category Sets A and B, capturing both rare and underperformed categories. The remaining categories are referred to as *common* categories.

**Gradient-Enhanced Underrepresented Class Matching (GEUCM)**

In the One-to-One matching scheme (Figure 2(a)), $m$ queries generate $m$ predicted masks, which are matched to $n$ ground-truth masks using a cost matrix that computes pairwise losses. The Hungarian algorithm then solves the resulting $m \times n$ assignment problem. The remaining $(m - n)$ queries (typically close to 100 vs. few food objects) are labeled "no object," introducing a large number of negative samples that dilute gradients for underrepresented classes. Although recent studies Li et al. (2024) have explored the One-to-Many matching (Figure 2(b)), assigning extra matches purely based on class frequency may degrade the performance of common classes, leading to suboptimal results in food segmentation.

To address this issue, we build on two key observations. First, rare classes contain few positive instances, and their gradients are easily overwhelmed by abundant negatives Li et al. (2024), increasing their match count strengthens their gradient signals and improves optimization balance. Second, queries naturally exhibit spatial biases Carion et al. (2020). For example, underrepresented classes in food images often occur in spatially predictable regions. Allocating dedicated queries to these classes promotes class-specific spatial priors, strengthening both their detectability and segmentability. Motivated by these insights, we propose Gradient-Enhanced Underrepresented Class Matching (GEUCM), a simple yet effective strategy that assigns extra matching queries for underrepresented categories. This enhances their gradient signals without compromising the learning of common categories, improving gradient distribution during training and ultimately boost overall segmentation performance.

Our approach consists of two steps. First, we apply One-to-One Matching for $m$ common queries, covering all ground-truth masks, comprising $n$ common and $u$ underrepresented masks. Second, we employ One-to-Many Matching (Figure 2(c)) for $e$ extra queries targeting the $u$ underrepresented classes, where each target is matched with $k$ queries, with $k$ defined as a hyperparameter. The matching strategies of these two steps operate independently, yielding a total of $m + u \times (k + 1)$ matched pairs. The full procedure is outlined in Algorithm 1.

---

**Algorithm 1** Gradient-Enhanced Underrepresented Class Matching

---

**Input**: A query set consisting of $m$ common queries for all ground-truth masks and $e$ extra queries for underrepresented ground-truth masks; a mask set containing $n$ common ground-truth masks and $u$ underrepresented ground-truth masks (for a single image inference).
**Parameter**: $k$ (number of extra matches per underrepresented ground-truth masks)
**Output**: $n + u \times (k + 1)$ matched query-mask pairs

1: **Initialize**: $\mathcal{M}_1 \leftarrow \emptyset$, $\mathcal{M}_2 \leftarrow \emptyset$
2: **Step 1: One-to-One Matching**
3: Compute a matching loss matrix between the $m$ common queries and all $(n + u)$ ground-truth masks.
4: Use the Hungarian algorithm to find the minimum-cost one-to-one assignment and store the results in $\mathcal{M}_1$.
5: **Step 2: Gradient-Enhanced One-to-Many Matching**
6: **for** t = 1 to $k$ **do**
7:    Compute a matching loss matrix between the remaining $e$ extra queries and $u$ underrepresented ground-truth masks.
8:    Apply the Hungarian algorithm to select a match per underrepresented ground-truth masks and update $\mathcal{M}_2$; remove the assigned queries from the extra query pool.
9: **end for**
10: **return** $\mathcal{M}_1 \cup \mathcal{M}_2$.

---

## 3 EXPERIMENTS

### 3.1 DATASETS

**FoodSeg103** Wu et al. (2021) is a large-scale dataset for food image semantic segmentation, containing 7118 high-quality images with pixel-level annotations across 103 common food categories. It presents challenges such as diverse categories, complex boundaries, and severe class imbalance, making it a strong benchmark for segmentation performance.

**UECFoodPixComplete** Okamoto & Yanai (2021) is a refined dataset for food classification and segmentation, comprising 10,000 images from 102 Japanese food categories. It improves upon the original UEC-FoodPix by manually correcting approximately 9,000 masks, addressing limitations of bounding box–based annotations and improving segmentation precision.

**Food50Seg** Aslan et al. (2020) includes 5,000 images across 50 food categories, with 100 pixel-level annotated images per class. It also provides 120,000 augmented images with various distortions, such as lighting variation, JPEG compression, and blur, enabling robustness evaluation.

## 3.2 EVALUATION METRICS

Following prior work, we evaluate model performance using mean Intersection over Union (mIoU), overall accuracy (aAcc), and mean accuracy (mAcc).

$$\text{mIoU} = \frac{1}{C}\sum_{i=1}^{C}\frac{\text{TP}_i}{\text{TP}_i + \text{FP}_i + \text{FN}_i}, \quad \text{aAcc} = \frac{\sum_{i=1}^{C}\text{TP}_i}{\sum_{i=1}^{C}(\text{TP}_i + \text{FN}_i)}, \quad \text{mAcc} = \frac{1}{C}\sum_{i=1}^{C}\frac{\text{TP}_i}{\text{TP}_i + \text{FN}_i}.$$

$$(9)$$

Here, $C$ represents the total number of classes, and the $\text{TP}_i$ denotes the number of pixels correctly classified as class $i$; $\text{FP}_i$ refers to the number of pixels incorrectly predicted as class $i$; and $\text{FN}_i$ represents the number of pixels whose true label is class $i$ but are misclassified as other classes.

| | Method | FoodSeg103 | | | UECFoodPixComplete | | |
|---|---|---|---|---|---|---|---|
| | | mIoU↑ | aAcc↑ | mAcc↑ | mIoU↑ | aAcc↑ | mAcc↑ |
| General Image Segmentation | FPNKirillov et al. (2019) | 27.28 | 75.23 | 36.7 | 53.34 | 83.93 | 67.21 |
| | DeeplabV3+Chen et al. (2018) | 31.04 | 79.32 | 42.66 | 55.50 | 66.80 | - |
| | CCNetHuang et al. (2019) | 28.6 | 78.90 | 47.80 | 64.62 | 87.96 | 77.50 |
| | UpernetXiao et al. (2018) | 39.8 | 82.02 | 52.37 | 59.35 | 86.64 | 74.44 |
| | SETRZheng et al. (2021) | 45.1 | 83.53 | 57.44 | 65.41 | 86.02 | 77.36 |
| | SegformerXie et al. (2021) | 38.67 | 80.29 | 48.96 | 64.89 | 88.25 | 76.98 |
| | KNetZhang et al. (2021a) | 40.18 | 82.01 | 52.01 | 64.88 | 87.63 | 76.94 |
| | FADCChen et al. (2024a) | 50.77 | 85.10 | 63.14 | - | - | - |
| | VimZhu et al. (2024) | 42.58 | 81.73 | 54.42 | 65.17 | 86.01 | 75.76 |
| | VMambaLiu et al. (2024) | 44.38 | 82.94 | 55.99 | 64.18 | 86.37 | 75.63 |
| | GCNetYang et al. (2025) | 46.25 | 83.94 | 56.21 | 65.33 | 87.46 | 76.73 |
| | SegMAN Fu et al. (2025) | 50.14 | 85.22 | 62.11 | 68.24 | 88.14 | 80.39 |
| Food Image Segmentation | STPPNWang et al. (2022) | 40.3 | 82.13 | 53.98 | - | - | - |
| | Window AttentionDong et al. (2021) | 31.4 | 77.62 | 40.3 | - | - | - |
| | ReLeM-CCNetWu et al. (2021) | 47.10 | 85.50 | 59.50 | - | - | |
| | FoodSAMLan et al. (2023) | 46.42 | 84.10 | 58.27 | 66.14 | 88.47 | 78.01 |
| | CANetDong et al. (2024) | 37.21 | - | 47.33 | 68.90 | - | 80.60 |
| | FDSNet Xiao et al. (2025) | 47.34 | - | 60.04 | - | - | - |
| | Mask2FormerCheng et al. (2022) | 51.37 | 86.40 | 64.39 | 68.39 | 89.34 | 78.68 |
| | **Ours** | **52.84** | **86.89** | **65.72** | **71.24** | **90.24** | **81.09** |

Table 1: Performance comparison of different methods on FoodSeg103(%) and UECFoodPixComplete(%). Best results are in bold, second-best are underlined.

## 3.3 IMPLEMENTATION DETAILS

All experiments are implemented in PyTorch and conducted on two NVIDIA RTX 3090 GPUs with a total batch size of 4 (2 images per GPU). The model is trained using the AdamW optimizer with an initial learning rate of 0.0001, following a cosine annealing schedule that gradually decays the learning rate to zero. $\alpha$, $k$, and $u$ are set to 0.5, 2, and 20, respectively, while the threshold $m$ is set to 0.3, 0.5, and 0.7 on the FoodSeg103, UECFoodPixComplete, and Food50Seg datasets, respectively. Unless otherwise specified, all hyperparameters are aligned with those in the Mask2Former baseline.

| | Method | Food50Seg | | |
|---|---|---|---|---|
| | | mIoU↑ | aAcc↑ | mAcc↑ |
| General Image Segmentation | DeeplabV3+Chen et al. (2018) | 70.01 | 84.94 | 81.26 |
| | CCNetHuang et al. (2019) | 74.25 | 82.56 | 81.36 |
| | UpernetXiao et al. (2018) | 71.29 | 83.65 | 82.59 |
| | SETRZheng et al. (2021) | 84.48 | 91.71 | 89.35 |
| | SegformerXie et al. (2021) | 85.14 | 90.94 | 88.56 |
| | KNetZhang et al. (2021a) | 83.74 | 92.27 | 90.18 |
| | VimZhu et al. (2024) | 82.95 | 88.85 | 90.63 |
| | VMambaLiu et al. (2024) | 84.18 | 91.11 | 90.92 |
| | GCNetYang et al. (2025) | 86.38 | 90.34 | 91.20 |
| | SegMAN Fu et al. (2025) | 89.57 | 95.48 | 95.37 |
| | Mask2FormerCheng et al. (2022) | 92.77 | 96.45 | 96.17 |
| | **Ours** | **93.42** | **96.78** | **97.02** |

Table 2: Performance comparison of different methods on Food50Seg(%).

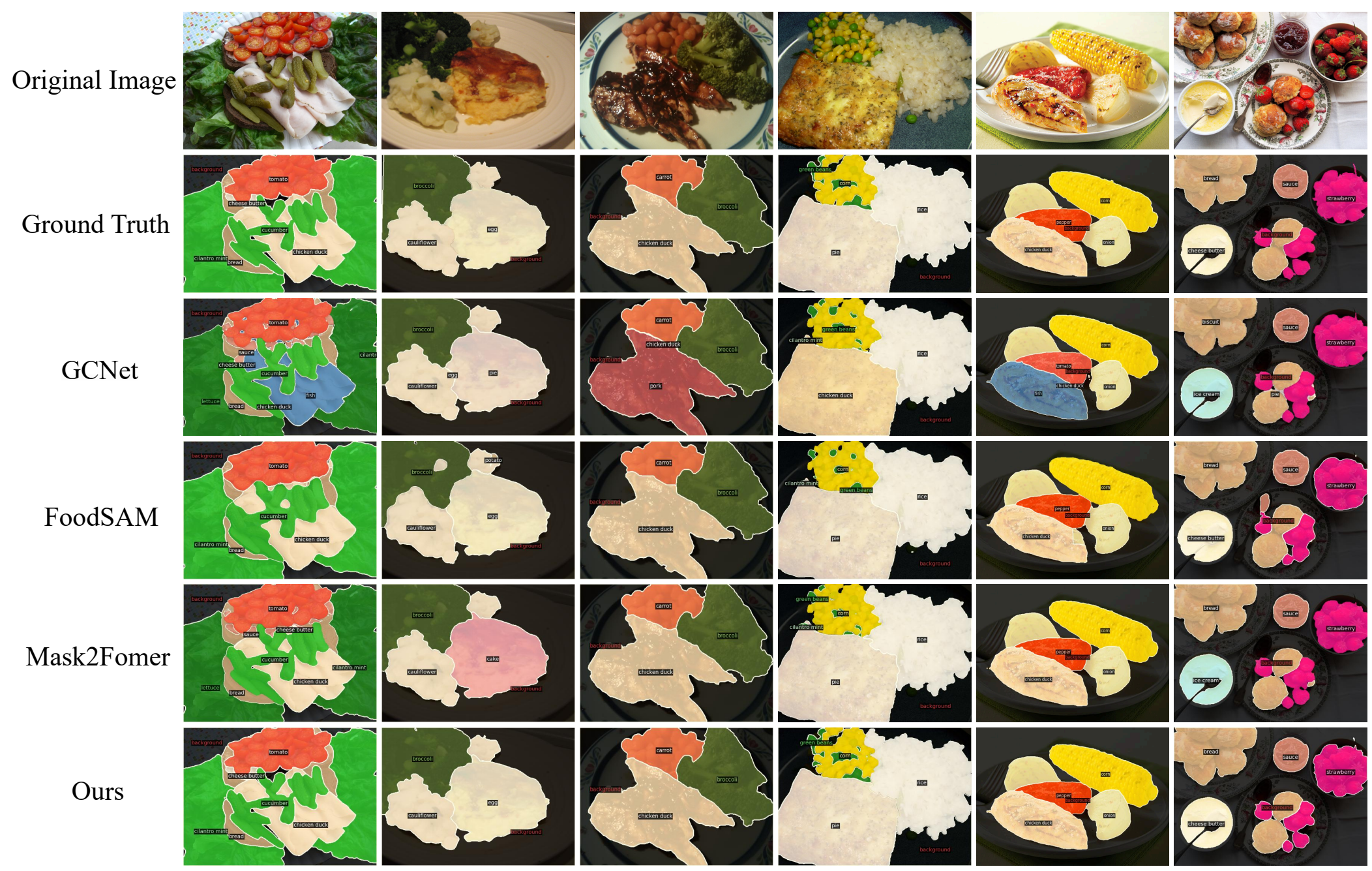

Figure 4: Visualization of food image segmentation across different segmentation frameworks, including FOCUS. FOCUS demonstrates better performance in terms of segmentation edge precision and mask classification accuracy..

### 3.4 Comparison Results

**Quantitative Comparison** We compare our proposed model against both recent SOTA general-purpose segmentation models and methods tailored to food image segmentation. As shown in Table 1 and Table 2, our method consistently outperforms all baselines, achieving new SOTA results across all datasets and metrics. While Mask2Former ranks as the second-best in most cases, our model surpasses it by a notable margin. Specifically, our model achieves mIoU scores of 71.24% on UECFoodPixComplete, 93.42% on Food50Seg, and 52.84% on FoodSeg103, representing a +1.47% improvement over Mask2Former on the most challenging dataset, FoodSeg103. This dataset presents a particularly challenging scenario due to its higher visual complexity and severe category imbalance, making these improvements even more significant.

**Qualitative Comparison** Figure 4 presents the qualitative comparison across different methods. Our approach demonstrates superior classification accuracy, particularly in challenging scenarios. In the first column, it accurately distinguishes overlapping components such as *cucumber* and *chicken*.

In the second column, our model correctly identifies *egg*, while GCNet and Mask2Former misclassify it as *pie* and *cake*, respectively. In the final column, our method successfully recognizes *Chinese butter* and precisely delineates the *spoon* within it, highlighting its strong capability in handling fine-grained targets and complex scenes.

### 3.5 ABLATION STUDY

In this section, we conduct ablation studies on FoodSeg103 to evaluate the effectiveness of the proposed components: FABM and GEUCM. As shown in Table 3a, both modules independently enhances model performance, and their combination yields the best results, showing clear synergy.

| FABM | GEUCM | mIoU↑ | aAcc↑ | mAcc↑ |
|------|-------|-------|-------|-------|
|      |       | 51.37 | 86.40 | 64.39 |
| ✓    |       | 52.49 | 86.54 | 65.24 |
|      | ✓     | 52.24 | 86.55 | 64.40 |
| ✓    | ✓     | **52.84** | **86.89** | **65.72** |

(a)

| Level1 | Level2 | mIoU↑ | aAcc↑ | mAcc↑ |
|--------|--------|-------|-------|-------|
|        |        | 51.37 | 86.40 | 64.39 |
| ✓      |        | 51.85 | 86.02 | 64.24 |
|        | ✓      | 51.95 | **86.70** | 64.11 |
| ✓      | ✓      | **52.49** | 86.54 | **65.24** |

(b)

Table 3: (a)Ablation study of different modules on FoodSeg103(%). (b)Ablation study of FDB on FoodSeg103 (%). Level1 correspond to output features from the first stage of the backbone, while Level2 represents the final output feature from the Pixel Decoder. The first row of the table data (without checkmarks) represents the baseline.

| $\alpha$ | mIoU↑ | aAcc↑ | mAcc↑ |
|----------|-------|-------|-------|
| 0.25     | 52.18 | 86.07 | 65.21 |
| 0.5      | **52.49** | **86.54** | **65.24** |
| 0.75     | 52.44 | 86.24 | 64.93 |

(a)

|                       | mIoU↑ | aAcc↑ | mAcc↑ |
|-----------------------|-------|-------|-------|
| w/o $L_{\text{FABM}}$ | 51.94 | 86.38 | 64.72 |
| w/ $L_{\text{FABM}}$  | **52.49** | **86.54** | **65.24** |

(b)

Table 4: (a) Ablation study of $\alpha$ on the FoodSeg103 (%). (b) Ablation study on the $L_{\text{FABM}}$ the FoodSeg103 (%).

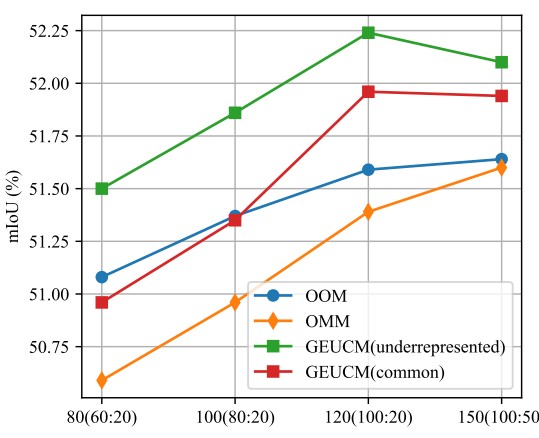

Figure 5: Comparison of different matching strategies in GEUCM on FoodSeg103 (%). The blue line denotes One-to-One Matching for all queries (OOM), the orange line denotes One-to-Many Matching for all queries (OMM), the green line represents GEUCM with extra queries for underrepresented classes, and the red line represents GEUCM with extra queries for common classes. The vertical axis indicates segmentation performance, while the horizontal axis shows the total number of queries. Numbers in parentheses specify the number of common and extra queries, respectively.

**Ablation of FDB in FABM** We evaluated the effectiveness of the Frequency Difference Block (FDB) in FABM by inserting it at different feature levels and analyzing its impact on segmentation performance. As shown in Table 3b, incorporating FDB at individual feature levels yields clear gains in both mIoU and aACC. Integrating FDB across all levels achieves the highest mIoU, the primary metric for segmentation, highlighting the benefit of full-level deployment.

**Choice of $\alpha$ in FABM** We perform an ablation study on the threshold $\alpha$ used for binary high–low frequency separation in FABM. As shown in Table 4a, the performance is not highly sensitive to the choice of $\alpha$. The results indicate that boundary refinement cues in food images are largely captured

| Method | mIoU | aAcc | mAcc |
|---|---|---|---|
| SegformerXie et al. (2021) | 50.10 | 82.36 | 61.77 |
| KNetZhang et al. (2021a) | 52.21 | 82.41 | 67.34 |
| VMambaLiu et al. (2024) | 51.64 | 81.23 | 66.85 |
| SegMANFu et al. (2025) | 53.21 | 83.77 | 67.84 |
| Mask2formerCheng et al. (2022) | 54.96 | 84.19 | 68.55 |
| **Ours** | **55.39** | **84.31** | **69.02** |

Table 5: Performance comparison of different methods on ADE20K(%).

by high-frequency components, thus, further subdividing the frequency spectrum would introduce unnecessary complexity without meaningful gains. Therefore, we set $\alpha$ to 0.5 in all experiments.

**Ablation of FABM Loss**  To validate the contribution of the FABM Loss, we conducted an ablation study that isolates this term. As shown in Table 4b, removing the $L_{\text{FABM}}$ leads to a clear drop in in multiple metrics, confirming the effectiveness of the edge-aware constraint.

**Choice of Matching Strategy in GEUCM**  As shown in Figure 5, our results demonstrate that, compared to other matching strategies, the proposed GEUCM(underrepresented) consistently delivers the best performance across various settings by introducing extra matching mechanisms exclusively for underrepresented classes. This highlights the effectiveness of our matching strategy.

Additionally, we observe that increasing the number of queries can somewhat enhance segmentation. However, this improvement is not guaranteed, when the proportion of extra matches becomes too large (i.e., an excessive number of underrepresented classes), performance may decline. This is likely because common classes are already optimized, offering limited room for further gains, and excessive attention may shift learning capacity away from the underrepresented classes.

### 3.6 COMPLEXITY ANALYSIS

We performed a comprehensive assessment of the computational efficiency of the proposed components. For FABM, both the FFT-based frequency decomposition and the multi-branch convolutional design are computationally lightweight, adding only approximately 4M parameters and 13 GFLOPs. For GEUCM, the additional cost primarily arises from the extra queries. With 20 extra queries (i.e., $u = 20$), GEUCM contributes fewer than 1M additional parameters and about 4 GFLOPs. In total, our method introduces roughly 5M additional parameters and 17 GFLOPs relative to the baseline, demonstrating that the performance gains are achieved with minimal computational overhead.

### 3.7 GENERALIZATION ANALYSIS

To assess our method's generalization, we perform additional experiments on ADE20K Zhou et al. (2017). We compare our method with the several SOTA methods. As shown in Table 5, our approach achieves mIoU, aAcc, and mAcc scores of 55.4, 84.3, and 69.0, respectively. Relative to the second-best method, Mask2Former, this represents absolute improvements of 0.4%, 0.1%, and 0.5%, demonstrating that our method generalizes well beyond the food segmentation domain.

## 4 CONCLUSION

In this study, we present FOCUS, a semantic segmentation framework tailored for food images that enhances both the mask proposal quality and mask classification accuracy. Specifically, we introduce a frequency-oriented edge enhancement module that sharpens boundary details during mask proposal generation, and a Gradient-Enhanced Underrepresented Class Matching strategy that assigns extra queries to underrepresented classes, thereby improving classification accuracy. Extensive evaluations on three benchmark food-image datasets demonstrate the effectiveness of our approach in advancing segmentation performance.

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
