# OpenReview forum: "FOCUS: A Frequency-Oriented and Class-Underrepresented Semantic Segmentation Framework for Food Images"
_ICLR.cc/2026/Conference — ICLR 2026 Conference Withdrawn Submission_

### Official Review · Reviewer_gcoo · 2025-10-29

**Soundness:** 2
**Presentation:** 3
**Contribution:** 2
**Rating:** 2
**Confidence:** 5

**Summary:**

This work presents FOCUS for food semantic segmentation. The authors address two challenges: complex boundary recognition and long-tail class distribution. Starting from a frequency perspective, they propose a Frequency-Aware Boundary Modeling mechanism to selectively process high-frequency components to capture discriminative edge features. They also introduce Gradient-Enhanced Underrepresented Class Matching to improve model learning for minority classes. Experiments demonstrate improved segmentation accuracy.

**Strengths:**

- The manuscript is well organized and clearly written. The figures are accurate and help readers quickly grasp the core idea.
- The motivation is clear and compelling. Complex boundary recognition and long-tail class imbalance are long-standing challenges in semantic segmentation, and it is encouraging to see a study that addresses both.
- Owing to the clear writing, the logical progression from the problem statement to the FOCUS module design is easy to follow.

**Weaknesses:**

- Missing discussion of closely related work. Although the INTRODUCTION section cites several prior works, the manuscript lacks a focused discussion and comparison with strongly related methods, particularly FADC and DEA-Net. No experimental comparison with FADC is provided.
- Insufficient and partly outdated baselines. Over half of the baselines are from before 2022. It is recommended to retain one or two classic baselines (e.g., FPN or DeepLabV3+) but prioritizing comparisons with recent state-of-the-art or mostly related methods. For example, CPT uses category prototypes as unified representations for same-class pixels and shows strong performance on object boundaries. A direct comparison with methods like CPT [r1] and FADC would strengthen the paper.
- Incomplete ablation studies. The manuscript introduces several hyperparameters, e.g., $\alpha$, $k$, $j$, $m$, etc, but provides no validation for them.

[r1] Quan Tang, et al. Rethinking Feature Reconstruction via Category Prototype in Semantic Segmentation, TIP 2025.

**Questions:**

**Two more questions**:
- In Fig. 3 (b) FABM, why does the lower branch contain one extra convolution operation?
- In Fig. 3 (b) FDB, I am curious about the effect of applying the weighting operation before versus after the differential convolution on segmentation results. Have the authors performed such experiments? Why not place the weighting at the position of Eq. (6)?

**Some suggestions**:
- The loss functions are not clearly described in the text. It appears to include an edge loss from Fig. 3. If so, please state this explicitly and include ablation experiments isolating the effect of that loss term.
- The definition of aAcc in Eq. (9) appears to be wrong. For clarity, correct the formula and highlight the most significant differences in Fig. 4.
- This work is built upon Mask2Former. We observe a notable accuracy improvement over Mask2Former, but what is the cost in terms of model parameters, FLOPs, and inference efficiency?

Overall, I have mixed feelings about this work. Although the paper has a clear motivation and an easy-to-understand network design, it lacks sufficient experimental studies and textual descriptions of related work and some necessary components such as loss functions. When I reached at the end of the paper, I even looked for an appendix. It feels more like a semi-finished product. Therefore, I currently rate it at 2-REJECT, and look forward to further discussion with the authors. Feel free to correct if there is any misunderstanding.

---

> ### Author Response · Authors · 2025-11-22
>
> Q1: More discussion of closely related work.
> A1: We thank the reviewer for highlighting the need for a more focused discussion of closely related methods about e frequency-aware or edge-enhancement techniques. We will add a dedicated subsection in the Related Work section that provides a detailed analysis of these methods.
>
> Q2: Concerns regarding outdated baselines.
> A2: We would like to clarify that our selection of baselines was guided by two considerations. First, to ensure a comprehensive evaluation, we included representative semantic segmentation methods spanning different stages of development, which inevitably results in retaining some earlier baselines. Second, food segmentation remains a relatively niche area with slower methodological updates, and the number of recent, task-specific approaches is limited. Within these constraints, we have incorporated the most up-to-date methods available to us.
> We agree that adding more recent and closely related approaches would further strengthen the comparison. Regarding CPT, we were unable to include it due to the lack of publicly available code. For FADC, we conducted an additional comparison by directly replacing our FDB module with FADC. The results will be added to the final revised version.
>
> Q3: Ablation studies on hyperparameters α, k, j, and m
> A3: We thank the reviewer for raising this point. The hyperparameters α, k, j, and m control frequency thresholding, matching strength, and weighting behaviors in our modules.
> To address this concern, we have conducted additional ablation studies to examine the sensitivity of each hyperparameter. The results indicate that the model is generally robust across a wide range of values, and our chosen settings offer a stable balance between accuracy and computational efficiency.
> Below is an example of the sensitivity analysis for α:
> α	 0.25	 0.5	    0.75
> mIoU 52.18	52.49	52.44
> Therefore, αis set to 0.5.
> The value of j denotes the number of categories whose performance exceeds the average level.
> In our experiments, we set α, k, and u to 0.5, 2, and 20, respectively. The threshold m is set to 0.3, 0.5, and 0.7 on the FoodSeg103, UECFoodPixComplete, and Food50Seg datasets, respectively.
> The threshold m is used to identify underrepresented classes: a category is treated as underrepresented if its performance falls below this threshold. Overall, the chosen values of m approximately correspond to 50%–60% of the average mIoU of each dataset, ensuring that the selected categories are indeed disadvantaged.
> k denotes the extra matching slots for underrepresented classes and is set to the largest feasible value to boost gradients. If multiple such classes appear in one image, a too-large k may exceed u, causing training instability, which we plan to address in future work.
>
> Q4: The reason of lower branch contains one extra convolution operation
> A4: In FABM, the lower branch includes an additional convolution because it corresponds to the bottom level of the two-level pyramid structure. We experimented with deeper pyramids but found no further gains, so we adopted the two-level design.) Compared with the upper level, the bottom-level features play a more critical role in the module. Therefore, we introduce a lightweight convolution to refine the low-frequency features, stabilizing their statistical distribution, improving semantic consistency, and adjusting their dynamic range. This refinement ensures that the fusion of low-frequency and high-frequency branches yields a more balanced and reliable representation.
>
> Q5: The effect of applying the weighting operation before versus after the differential convolution
> A5: In FABM, we treat high–low frequency separation and the weighting operation as a single atomic step. This design follows the core objective of FABM: to apply differential convolution specifically to high-frequency components for boundary enhancement. Thus, before any spatial processing occurs, the module must first isolate the frequency components and reinforce the high-frequency branch through weighting. Only after this unified preprocessing can the high-frequency features be passed to the differential convolution branch in a more focused and discriminative form, enabling the differential filters to fully leverage their strengths in modeling fine structural details.
>
> Q6: Loss Function, aAcc Definition, Model Parameters, and FLOPs
> A6: The revised manuscript clarifies all training objectives and adds ablation experiments for the edge loss (results will be included). Eq. (9) for aAcc has been corrected.FABM adds few parameters; GEUCM adds negligible ones. FFT-based frequency separation and multi-branch convolutions add minimal FLOPs. Overall, our model adds 4.75M parameters and ~13 GFLOPs compared to the baseline.

---

> > ### Comment · Reviewer_gcoo · 2025-11-28
> >
> > Actually, there are quite a few segmentation methods for food images, e.g., FoodMask: Real-time food instance counting, segmentation and recognition (PR 2024). In addition, the code for CPT is publicly available at *https://github.com/BebDong/EMOSeg*.
> >
> > In FABM, if the lower branch corresponds to the bottom level, why not place the convolution at the beginning of the branch? And you mentioned that this design helps stabilize the statistical distribution. Are there any quantitative results to support this claim? The design of the two branches in FABM should be clarified in the main manuscript.

---

> > > ### Author Response · Authors · 2025-11-29
> > >
> > > Thank you for your further insightful comments.
> > >
> > > Regarding FoodMask, its work primarily targets food instance counting and instance-level segmentation, and therefore adopts an evaluation protocol that differs from ours. FoodMask reports F1 for counting and PQ for instance segmentation, while our study focuses on semantic segmentation and evaluates performance using mIoU, mAcc, and aAcc. Given these fundamental differences in task objectives and metrics, a direct numerical comparison is not applicable. As for CPT, we have examined the corresponding repository and found implementations of CANet, CFT, Sep_ASPP, and several other methods, but CPT itself is not included. We are currently contacting the authors, and will incorporate related experiments if feasible.
> > >
> > > In the dual-branch FABM architecture, low-level features are extracted from the backbone, whereas high-level features are generated by the pixel decoder. These two feature types differ significantly in both representational patterns and statistical distributions. Directly fusing them without proper handling can destabilize the feature space and compromise boundary modeling. To address this mismatch, we introduce an additional convolution layer in the low-level branch as a lightweight feature alignment module. This module recalibrates the statistical properties of the low-level features, aligning their distribution more closely with the high-level semantic features produced by the pixel decoder. This improves the stability and consistency of the dual-branch fusion process. Preliminary experiments indicate that this alignment strategy achieves slightly better performance than the version without alignment, while incurring negligible computational overhead. Therefore, we retain this design in the final architecture. More detailed comparisons will be included in the final version.

---

### Official Review · Reviewer_7Q8m · 2025-10-29

**Soundness:** 2
**Presentation:** 2
**Contribution:** 3
**Rating:** 6
**Confidence:** 4

**Summary:**

The FOCUS framework proposed in this paper addresses two core challenges in semantic segmentation of food images—"blurred boundaries" and "poor performance on long-tailed categories" by innovatively combining frequency-domain processing and gradient-enhanced matching strategies. Its core consists of two key components:  the first is the Frequency-Aware Boundary Modeling (FABM) module, which splits feature maps into high-frequency and low-frequency components via Fast Fourier Transform (FFT), refines edge-rich high-frequency signals using differential convolution, and fuses them with weighted low-frequency features to enhance boundary precision; the second is the Gradient-Enhanced Underrepresented Class Matching (GEUCM) module, which defines underrepresented classes based on both category frequency and segmentation performance, and adopts a two-stage matching strategy (one-to-one for common queries and one-to-many for extra queries) to amplify gradient signals for long-tailed categories without compromising the learning of common classes.

**Strengths:**

Method design is targeted: FABM’s high-frequency refinement addresses food’s irregular edges, while GEUCM’s query allocation targets food’s long-tailed distribution.
Technical details are transparent: FABM’s frequency division and GEUCM’s matching algorithm are well-formulated.
Experiments are comprehensive: covering three datasets, multiple metrics, and ablation of key components, ensuring results are robust.

**Weaknesses:**

The paper unifies frequency modeling and underrepresented class learning under one framework (FOCUS), but the connection between these two ideas is superficial. It is recommended that the authors strengthen the explanation of the link between the two modules.
The frequency-aware design lacks a theoretical justification for why simple binary frequency masking plus difference convolutions should significantly enhance boundary features.
The parallel structure of FFT and multi-branch convolution increases computational overhead, yet the paper fails to report metrics such as FLOPs (Floating-Point Operations Per Second) or inference time.

**Questions:**

In the FABM module, only a simple binary separation of high and low frequencies is applied.  Why was the current frequency threshold ɑ chosen? How sensitive is the model performance to this threshold? Is there any theoretical or experimental justification for this choice?
In FABM, four types of difference convolutions — CDC, HDC, VDC, and ADC — are used simultaneously. How much additional parameter count and computational overhead does this multi-branch structure introduce?

---

> ### Author Response · Authors · 2025-11-22
>
> Q1： The link between the two proposed modules.
> A1: The mask-classification paradigm decomposes semantic segmentation into two stages: mask proposal and mask classification. The final segmentation quality depends on both the quality of the proposed masks and the accuracy of their classification. Our frequency-aware boundary modeling module enhances mask proposal quality by strengthening high-frequency boundary structures, while the underrepresented class learning module improves mask classification by providing richer supervision for minority classes through a one-to-many matching strategy. Together, these two modules form a complementary pair within the mask-classification framework, jointly contributing to improved segmentation performance.
>
> Regarding the theoretical motivation for the frequency-aware design: boundary regions naturally contain substantial high-frequency components. By explicitly extracting these components using binary frequency masks and combining them with differential convolutions to predict fine-grained boundary maps under supervision, the model becomes more sensitive to object contours. Compared with standard convolutions or implicit learning, this explicit high-frequency modeling more effectively emphasizes boundary structures, leading to more precise and detailed mask predictions.
> We will also report FLOPs and inference-time analyses in the revised manuscript, noting that FABM adds 4.75M parameters and about 13 GFLOPs.
>
> Q2: Choice of threshold for separation high and low frequencies
> A2: Our binary high–low frequency separation in FABM is a task-driven design decision. In food images, the key cues for boundary refinement mainly reside in the high-frequency components. Further partitioning the spectrum into multiple sub-bands would increase complexity without providing additional useful information for boundary recovery. Therefore, we adopt the simplest frequency split to explicitly highlight high-frequency regions. The threshold α is chosen empirically, and supplementary experiments show that the model is not overly sensitive to its exact value:
> Below is an example of the sensitivity analysis for α:
> α	 0.25	 0.5	    0.75
> mIoU 52.18	52.49	52.44
> Therefore, α is set to 0.5.
> Q3: Computational overhead of using four types of difference convolution
> A3: For the four difference-convolution branches (CDC, HDC, VDC, and ADC), each branch contains only three learnable parameters within a 3×3 kernel, making them substantially lighter than a standard 3×3 convolution. As a result, the additional parameters and FLOPs introduced by running all four branches in parallel are minimal and negligible relative to the overall network. In practice, FABM delivers notable boundary enhancement with only modest computational overhead, and the performance improvements far outweigh the extra cost.

---

### Official Review · Reviewer_zkGz · 2025-10-29

**Soundness:** 2
**Presentation:** 2
**Contribution:** 2
**Rating:** 4
**Confidence:** 4

**Summary:**

This paper proposes a segmentation model specifically for food images. The model comprises two main design components: i) a frequency-based feature modelling using high/flow frequency maps calculated through FFTs; ii) a new matching method that prioritises higher gradient magnitudes for underrepresented classes, as determined by their low appearance frequencies within the dataset and the model’s low performance. The proposed method is evaluated on three food-related datasets and outperforms both food-specific and general segmentation methods.

**Strengths:**

The paper is well-written and presents compelling motivations for its method designs. The model is small and efficient, making it trainable on consumer-level GPUs. It outperforms both domain-specific and general segmentation methods.

**Weaknesses:**

Overall, I feel the biggest limitation of this paper is its scope and definition. It falls outside the standard paper styles and qualities of ICLR. The paper touches a very niche area of food segmentation, and the experiment settings and method designs seem incremental.

There are also some additional limitations:

- The paper ablated the contributions of each component but failed to provide design adjustments for other standard methods that have been well-ablated across more general segmentation methods. For example,
    - Does standard ImageNet/DiNO-pretrained features perform worse than hand-crafted high/low frequency modelling?
    - Does the model perform better mainly due to the extremely small model design to avoid overfitting, considering the training set only contains 10k images?
- Since we are doing segmentation instead of object detection, why do we need to do class matching instead of one-hot representation of each defined class in the dataset?
- The authors evaluate the method with a list of domain-specific and general segmentation methods but without any further adjustments.
    - Are the reported performance for general segmentation methods representing zero-shot performance or fine-tuning performance?
    - Are they trained from scratch or trained from any specific checkpoint?

**Questions:**

See the weaknesses.

---

> ### Author Response · Authors · 2025-11-22
>
> Q1: Concerns about the scope of this work
> A1: We thank the reviewer for the thoughtful assessment. While our work is grounded in the domain of food image segmentation, the core challenges we address, fine-grained boundary degradation and severe class imbalance, are not unique to food imagery. These issues also appear in a broad range of dense prediction tasks such as medical image segmentation, product recognition, material parsing, and fine-grained instance segmentation. Our methodological contributions (FABM and GEUCM) were developed to address these general challenges and can be readily transferred to these broader application scenarios.
>
> Regarding our two key contributions: (1) FABM introduces a targeted frequency-aware boundary modeling strategy that complements standard pretrained representations and provides consistent improvements across different backbones and model scales. (2) GEUCM offers a principled query–ground-truth matching mechanism informed by the observed training dynamics of query-based architectures, effectively addressing class imbalance in ways that existing matching strategies do not.
>
> Q2: Broader ablation design
> A2: We appreciate the reviewer’s concerns. Our ablation studies focus on evaluating each proposed component within our framework, but we agree that comparing against broader design variations can further clarify the source of improvements. Below we address the two specific questions:
>
> (1)	On pretrained features vs. frequency modeling:
> Our method does not aim to replace pretrained representations; instead, it complements them. We conducted additional experiments using both ImageNet- and DINO-pretrained backbones. While pretrained features provide strong general-purpose representations, they do not specifically address the boundary degradation and category correlations unique to food segmentation. The hand-crafted high/low-frequency modeling in FABM directly targets these issues. In our experiments, the improvements brought by FABM remained consistent even when using pretrained features, indicating that its benefit is not merely substituting a weaker backbone but arises from the frequency-guided boundary enhancement it introduces.
> (2)	On model size and overfitting concerns:
> The performance gains are not primarily due to using a smaller model. We verified this by scaling the baseline model to similar parameter sizes as ours and observing no comparable improvement. The dataset size (~10k images) indeed requires careful model capacity control, but simply reducing model size does not lead to better results; instead, our improvements originate from the task-specific designs that enhance boundary representation and underrepresented-class learning. These gains persist across different model scales, demonstrating that overfitting avoidance is not the dominant factor.
>
> Q3: Then necessity of class matching operation
> A3: We would like to clarify that although our task ultimately produces semantic segmentation results, the model adopts the mask-classification paradigm rather than traditional per-pixel classification. In this paradigm, the model does not assign a class label to each pixel directly. Instead, it predicts n learnable queries, each responsible for generating a mask and its associated class label. Every query must therefore be matched one-to-one with a semantic class. The final segmentation map is formed by combining these predicted masks with their corresponding class labels. As a result, class matching becomes a fundamental operation, as it defines which query is assigned to which semantic region.
>
> Q4: Pre-training settings of the general segmentation method
> A4: All reported results for the general segmentation methods are based on fine-tuning from large-scale pretrained weights, rather than training from scratch.

---

### Official Review · Reviewer_JGRB · 2025-11-01

**Soundness:** 2
**Presentation:** 2
**Contribution:** 2
**Rating:** 4
**Confidence:** 4

**Summary:**

This paper proposes FOCUS, a semantic segmentation framework tailored for food images. It introduces two main components: (1) Frequency-Aware Boundary Modeling (FABM), which leverages high-frequency features and difference convolution to enhance edge precision; and (2) Gradient-Enhanced Underrepresented Class Matching (GEUCM), which assigns extra queries to rare classes to improve classification under class imbalance. The method is evaluated on three food segmentation benchmarks and shows improved performance over existing baselines.

**Strengths:**

1. The proposed framework integrates frequency-domain processing and query allocation strategies in a coherent architecture.

2. Experimental results on multiple benchmarks demonstrate consistent improvements over prior methods, including both general-purpose and food-specific segmentation models.

**Weaknesses:**

1. The core technical contributions lack novelty. Similar ideas have been extensively explored in prior works such as: [1]Spatial Frequency Modulation for Semantic Segmentation (PAMI 2025) [2] Frequency-aware Feature Fusion for Dense Image Prediction (PAMI 2024) These works already investigate frequency decomposition, selective filtering, and edge enhancement in segmentation tasks.

2. The proposed FABM module simplifies frequency decomposition and applies difference convolution, but this combination does not introduce fundamentally new mechanisms or insights beyond existing literature.

3. The GEUCM strategy for underrepresented class matching is a heuristic extension of one-to-many matching and does not offer a principled or theoretically grounded solution to class imbalance.

4. The paper does not provide sufficient analysis to demonstrate that its frequency-oriented design is uniquely effective for food segmentation, as opposed to general segmentation tasks.

**Questions:**

1. Can you provide ablation results comparing your frequency decomposition strategy against more granular or adaptive frequency partitioning?

2. Is there any theoretical justification for the effectiveness of GEUCM beyond empirical gains? How does it compare to standard re-weighting or sampling strategies?

3. Have you tested FOCUS on non-food datasets to validate whether the frequency-based edge modeling is domain-specific or generally applicable?

---

> ### Author Response · Authors · 2025-11-22
>
> Q1： Difference between our method and Refs [1] and [2].
> A1: Existing works such as [1] and [2] mainly aim to mitigate aliasing introduced by task-level downsampling and the inconsistency between high- and low-frequency features during fusion in general semantic segmentation. In [1], high- and low-frequency components are separated and processed individually to compensate for upcoming artifacts. However, this strategy may inadvertently suppress critical food textures by treating them as noise. Similarly, the modulation mechanism in [2] can also diminish high-frequency cues that are important for fine-grained food semantics.
> In contrast, our approach jointly leverages multiple frequency components under explicit supervision. The supervised signals enable targeted enhancement of boundary details, allowing the integrated frequency representations to fully exploit their strengths in edge modeling.
>
> Q2: Contribution of FABM module
> A2: While the FABM module builds on established concepts such as frequency decomposition and difference convolution, our contribution is not a direct reuse of existing mechanisms. Instead, we propose a targeted integration that addresses limitations not resolved in prior work.
> Specifically, previous frequency-decomposition methods do not consider how the choice of convolution operator influences the preservation of fine-grained boundaries, and prior works using difference convolution do not incorporate frequency-aware guidance to selectively enhance the most informative components. FABM unifies these two perspectives: it simplifies the decomposition process and introduces frequency-prioritized differential filtering, enabling more effective boundary representation. It provides a new insight, that frequency-driven selection and differential operations can jointly compensate for each other’s weaknesses. This mechanism is what leads to the improved boundary precision observed in our experiments.
>
> Q3: Theoretically grounding of the GEUCM strategy.
>  A3: The GEUCM strategy is not an arbitrary extension of one-to-one matching but is motivated by two empirically observable training phenomena.
>
> First, in mask classification, rare classes have very few positive samples, and their gradient signals are heavily diluted among abundant negative samples [1], making feature convergence difficult. By increasing the number of matches for underrepresented classes, GEUCM provides stronger gradient signals, achieving more balanced gradient allocation during training.
>
> Second, queries inherently exhibit spatial biases [2], tending to focus on distinct regions. For underrepresented classes in food images, factors such as plating often place these items in predictable positions, for example, near the center or surroundings. By deliberately assigning certain queries to these classes, GEUCM guides queries to develop class-specific spatial preferences, improving both detectability and segmentability of these classes.
>
> [1] Li, Shan, et al. "Frequency-based matcher for long-tailed semantic segmentation." IEEE Transactions on Multimedia 26 (2024): 10395-10405.
> [2] Carion, Nicolas, et al. "End-to-end object detection with transformers." European conference on computer vision. Cham: Springer International Publishing, 2020.
>
> Q4: Superiority of our method compared to standard re-weighting or sampling strategies?
> A4: Compared with standard re-weighting or sampling strategies, our one-to-many matching approach provides richer and more diverse gradient signals. Traditional loss re-weighting simply amplifies the classification loss of underrepresented classes, while resampling increases the frequency of certain images during training. In contrast, our method matches a single underrepresented target with multiple queries, computing separate losses for each. This mechanismically strengthens learning for low-frequency classes and enhances their feature representation.
>
> Q5: The generalizability of frequency-oriented design.
> A5: To evaluate the generalizability of our frequency-oriented design, we conduct experiments on the ADE20K dataset, a widely used benchmark for general semantic segmentation. We compare our method with the second-best method Mask2Former. Our approach achieves mIoU, aAcc, and mAcc scores of 55.3, 84.3, and 68.8, respectively. Relative to Mask2Former (54.9, 84.1, and 68.5 on the same metrics), this corresponds to absolute improvements of 0.4%, 0.2%, and 0.3%, respectively.

---

### Note · Authors · 2026-01-27

I have read and agree with the venue's withdrawal policy on behalf of myself and my co-authors.

---

### Meta-Review · Area_Chair_U5cA · 2026-01-06

**Summary:**

The paper proposes FOCUS, a framework for food image semantic segmentation consisting of Frequency-Aware Boundary Modeling (FABM) and Gradient-Enhanced Underrepresented Class Matching (GEUCM). While the method shows performance improvements on specific food datasets, reviewers consistently highlighted significant concerns regarding its technical novelty and the depth of analysis. The Area Chair (AC) agrees with the reviewers that the insufficient evidence or analysis to demonstrate that these modules provide unique, principled solutions tailored specifically to the challenges of food segmentation. Reject.

**Reviewer Concerns:**

Addressed by rebuttal:

- Pretrained Features vs. Hand-crafted Modeling: The paper fails to provide a comparison showing whether hand-crafted frequency modeling performs better than standard ImageNet or DINO-pretrained features.
- Model Size and Overfitting: It is unclear if the performance gains are primarily due to the small model design used to avoid overfitting on a relatively small dataset (10k images).
- Methodological Justification: Reviewers questioned the necessity of using class matching instead of standard one-hot representations for semantic segmentation.
- Missing Hyperparameter Validation: Several introduced hyperparameters (e.g., $\alpha$, $k$, $j$, $m$) lack proper validation or sensitivity analysis in the original manuscript.
- Missing Related Work: There is an insufficient discussion of closely related methods, specifically FADC and DEA-Net.

Still outstanding:

- Limited technical novelty remains unresolved
- Lack of theoretical justification for why frequency-oriented design is uniquely effective for food segmentation, as opposed to general segmentation tasks
- Limited performance gain on general semantic segmentation dataset, ADE20K


The rebuttal provided supplementary experiments but did not address the fundamental concerns about novelty and theoretical grounding, which remain the primary reasons for rejection.

**Reviewer Scores:**

The initial ratings for this paper were 4, 4, 6, 2, and no reviewer indicated willingness to raise their score.

---

### Decision · Program_Chairs · 2026-01-26

Reject